# Is teenage parenthood associated with early use of disability pension? Evidence from a longitudinal study

**Fredinah Namatovu**[1,2]* , **Erling Häggström Gunfridsson**[2] , **Lotta Vikström**[2,3]

**1** Department of Epidemiology and Global Health, Umeå University, Umeå, Sweden, **2** Centre for Demographic and Ageing Research (CEDAR), Umeå University, Umeå, Sweden, **3** Department of Historical, Philosophical and Religious Studies, Umeå University, Umeå, Sweden

These authors contributed equally to this work.
* fredinah.namatovu@umu.se

**Data Availability Statement:** We do not have permission to share the data. Data is owned by statistics sweden (SEB), so we cannot make this dataset publicly available due to ethical and legal restrictions regarding the Swedish Public Access to

## Abstract

### Background

Over the past decades the number of young people using disability pensions (DP) has gradually increased in Europe but the reasons for this change are poorly understood. We hypothesize that teenage parenthood could be associated with an increased risk of receiving early DP. The aim of this study was to examine the association between having a first child at age 13–19 and receiving DP at age 20–42 (here called early DP).

### Methods

A longitudinal cohort study was undertaken based on national register data obtained from 410,172 individuals born in Sweden in 1968, 1969, and 1970. Teenage mothers and fathers were followed until age 42 and compared to non-teenage parent counterparts to examine their early receipt of DP. Descriptive analysis, Kaplan-Meier curves, and Cox regression analyses were performed.

### Results

The proportion of teenage parents was more than twice higher in the group that received early DP (16%) compared to the group that did not receive early DP (6%) during the study duration. A higher proportion of teenage mothers and fathers started to receive DP at 20–42 years old compared to non-teenage parents, and the difference between the two groups increased during the observation period. A strong association was observed between being a teenage parent and receiving early DP, significant both independently and after adjusting for the year of birth and the father's level of education. From the age of 30 to 42 years, teenage mothers used early DP more often than teenage fathers or non-teenage parents, and this difference also increased during the follow-up period.

Information and Secrecy Act. Interested researchers can access this dataset through contact with Statistics Sweden, a Swedish government agency responsible for official statistics in Sweden. To request data access from this agency contact; information@scb.se and +46104795000. Interested researchers should also consider obtaining ethical approval from the Swedish Central Ethical Review Board, its contact below: registrator@etikprovning.se, telephone: +46104750800

**Funding:** The salaries for FN, EL and LV's to contribute to this study is financed by the DISTIME project headed by Lotta Vikström, that has received funding from the Marcus and Amalia Wallenberg Foundation, MAW 2019.0003), "Ageing with disabilities: Risks and loads from disabilities and later life outcomes". The funders had no role in the study design, data collection, data analysis, data interpretation, or writing of the report.

**Competing interests:** The authors have declared that no competing interests exist.

## Conclusion

A strong association was found between teenage parenthood and the use of DP between 20 and 42 years of age. Teenage mothers used DP more than teenage fathers and non-teenage parents.

## Introduction

Disability pension (DP) is considered an important part of the public support programs for people with disabilities, it is a salary replacement for people of working age with long-term health limitations preventing them from working [1]. Several countries in Europe have witnessed an increase in the proportion of young adult recipients of DP [1–3], which has prompted inquiries into the associated risk factors [3–7]. The life course approach is used to explore how various factors through life influence health and the subsequent use of DP [5–8]. This approach asserts that exposure to certain risk factors at different periods of growth and development in life has a long-term influence on the development of chronic diseases [9]. However, research on risk factors for DP has mainly focused on conditions occurring during the perinatal period [5, 7, 10–13] and adulthood [4, 14, 15], and not during the teenage period.

The teenage period is an important formative phase in life when major physical, emotional, and social changes happen, and identity formation begins to take place. According to WHO, globally 1 in 7 (14%) of 10–19-year-olds experience mental ill-health, accounting for 13% of the global burden of disease within this age group [16]. A systematic review found a high prevalence of mental ill-health (in particular, depression) among adolescents after key sexual and reproductive health events such as pregnancy, abortion, or childbirth [17].

Parenthood, even when desired and planned, can be complex and demanding and at times can negatively affect one's overall health and more so mental health, manifesting as stress, anxiety, and depression in mothers and fathers regardless of age [18]. A well-established body of research shows that teenage parents are at an increased risk for adverse health and socio-economic outcomes later in life [19–21]. Moreover, some studies show that teenage mothers are more likely to report poor mental health in adulthood than women who did not have children as teenagers [19, 22–24]. Other studies show that teenage mothers have poorer self-reported health later in life, poor socio-economic outcomes later in life, and poorer maternal, physical, and mental health outcomes than do non-teenage mothers [19, 25, 26]. Available reports on teenage fathers also reveal high levels of anxiety and depression in this group [27, 28]. A systematic review of 18 studies on fathers shows an increase in paternal stress during the perinatal period and around the time of birth [29].

Interestingly, several psychiatric diagnoses are identified as strong predictors for later use of DP [4, 30, 31]. For example, a Swedish study shows that DP is used significantly more often by those with psychiatric diagnoses, most particularly for women under 30 years of age [4]. Another Swedish study also reported a high incidence of DP among persons aged 16–64 years with a psychiatric diagnosis [30]. A Norwegian study demonstrated a strong association between mental illness and disability retirement [31].

Based on the observation that psychiatric diagnoses are a common route to early use of DP, we hypothesized that teenage parenthood would be associated with an increased risk of receiving DP in young adulthood. Up to now, no longitudinal studies that we know of have attempted to investigate this association among men and women in Sweden. Thus, we wanted to clarify whether teenage parenthood is associated with early DP in Sweden.

### Study aim

The aim of this study was to examine the association between teenage parenthood and the use of DP in young adulthood and to assess whether there were differences in early DP use between men and women.

## Materials and methods

This longitudinal cohort study is based on national register data obtained from 410,172 individuals born in Sweden in 1968, 1969, and 1970. These birth cohorts were chosen because they were the oldest in our data set and could provide sufficient follow-up duration. The original total population was 440,220 persons, but 30,048 (6.8%) were excluded from the analysis because of their death or emigration before age 20 or because the information was missing about sex, or if their personal identification number had been reused.

The main outcome variable was DP status represented by the "age at first receiving DP". This age ranged between 20 and 42 years, the age range was chosen due to data availability. Those who received DP at age 20 could have started receiving DP earlier than this, however, we considered the age that was first recorded in the Longitudinal Integration Database for Health Insurance and Labour Market Studies (LISA) database following its establishment in 1990. The highest cut-off age was based on data available in the SIMSAM Lab, data was only available up to 2010 when the oldest birth cohort was aged 42 years. The DP is a financial security program developed to offer financial relief to people with reduced working capacity due to having a chronically disabling health condition [1, 2]. Eligibility for receipt of DP is confirmed through a medical examination which indicates diminished health and work capacity [1]. We coded individuals who received DP as "yes" in the first year they received DP; those who did not receive DP in the study's timeframe were coded "no". The predictor variable under study was whether an individual had been a teenage parent (i.e., they had become parents at 19 years or younger) or not. Other covariates included in this study were sex, year of birth, and father's highest level of education obtained. Father's level of education was used due to data availability, it was grouped as the university, secondary, primary, and missing (for those who lacked data on education).

All data was obtained from Statistics Sweden (SCB). Data on DP status was obtained from the LISA database. Data on the age at birth of the first child was obtained from the total population register and the multi-generation register while data on sex and year of birth were obtained from the total population register. Information on the index father's education level was obtained from the 1970 Population and housing census. All data from the several registers were linked by Statistics Sweden (SCB) using the personal identification number. Linked data was anonymized by Statistics Sweden before it was made available for this analysis through the Swedish Initiative for Research on Microdata in Social and Medical Sciences (Umeå SIMSAM Lab) [32].

### Statistical analyses

Descriptive statistics that give an overview of the data are presented as frequencies. To explore the association between age at having a first child and whether a DP was received in early adulthood, we used Kaplan-Meier curves and Cox regressions. We plotted the Kaplan-Meier curves to examine patterns in the proportion that received DP between ages 20 to 42 by teenage parent status and sex.

Using Cox proportional hazard regression, we modeled having DP during the follow-up duration by comparing teenage parents to individuals who were not teenage parents. Individuals were followed from age 20 until they received a DP or until they were censored, due to out

migration, death, or by the end of study period on 31 December 2010. The independent and adjusted associations for men and women are presented as hazard ratios (HR) with 95% confidence intervals and statistical significance set at p < 0.05.

All analyses were performed in the statistical program environment R version 4.0.2 [33].

### Ethics approval

The Regional Ethical Board has approved all research based on Swedish population data accessed through the Umeå SIMSAM Lab, including the present study (Dnr: 2010-157-31 Ö). Statistics Sweden made all the data anonymous before making it available for research, therefore obtaining informed consent for each individual was neither possible nor necessary.

### Results

Analysis of the overall demographic information (Table 1) indicates that, in the group that received DP, the proportion who had been teenage parents (15%) was more than twice higher than those who had not been teenage parents (6%). Teenage parenthood was more common among women who received DP (16%) than men who received DP (9%).

As shown by the Kaplan-Meier curves in Fig 1, a higher proportion of teenage mothers and fathers started to receive DP earlier than non-teenage parents, and this difference between the two groups increased as the cohort aged. It can also be clearly seen that teenage mothers were more likely to have received an early DP than teenage fathers.

Table 2 shows that the association between teenage parenthood and the use of DP was statistically significant both independently and after adjusting for year of birth and father's educational level, although the chances of having an early DP did not differ by sex.

### Discussion

Our study found a positive significant association between being a teenage parent and receiving DP during early adulthood. The Kaplan-Meier curves show large differences between the sexes in the use of DP. From age 30 up to 42, there was a widening gap in the use of DP, with a more rapid increase in the use of DP among teenage parents of both sexes compared to non-

**Table 1. Demographic characteristics of men and women born between 1968–1970 (N = 410,172), by DP status.**

| Characteristics | | No DP (n = 416 003) | DP (24 218) |
|---|---|---|---|
| | | N (%) | N (%) |
| **Teen parent** | Yes | 8,503 (85) | 1,460 (15) |
| | No | 377,893 (94) | 22,226 (6) |
| **Female teen parent** | Yes | 7,047 (84) | 1,308 (16) |
| | No | 179,471 (93) | 13,480 (7) |
| **Male teen parent** | Yes | 1,456 (91) | 152 (9) |
| | No | 198,512 (96) | 8,746 (4) |
| **Year of birth** | 1968 | 131,529 (94) | 8,677 (6) |
| | 1969 | 126,347 (94) | 7,835 (6) |
| | 1970 | 128,610 (95) | 7,174 (5) |
| **Father's highest education level** | University | 32,678 (96) | 1,231 (4) |
| | Secondary | 113,289 (94) | 6,638 (6) |
| | Compulsory | 138,215 (93) | 10,828 (7) |
| | Missing | 102,304 (95) | 4,989 (5) |

teenage parents. The Kaplan-Meier curves also showed that teenage mothers were more likely to receive DP than teenage fathers and those who were not teenage parents. The Kaplan-Meier curves were corroborated by the Cox regressions which showed high hazard ratios for teenage parents to receive early DP. These hazard ratios are at a similar level for both sexes, which is in line with the more rapid increase for females using DP because the baseline risk for females is higher and the hazard ratios act as multipliers of that baseline risk. The Cox regression also shows that the probability of using DP was higher among index persons whose fathers had at most a secondary or compulsory education compared to those whose fathers had had a university education and in persons born in 1969 and 1970 compared to the birth cohort of 1968.

Our finding that having a first child as a teenager was associated with a high probability of starting to use DP during early adulthood, suggests that teenage parenthood contributes to later adverse health outcomes [22, 23]. One possible explanation for this increased risk in early DP use among former teenage parents is that having a child during the teenage years (a critical stage in human development) forces an accelerated psychological transition to adulthood, which induces stress and its consequent negative health effects [34, 35]. A British study published in 2002 showed more mental health problems and socio-economic deprivation among teenage mothers born in the 1980s and 1990s than had been observed in earlier birth cohorts [36]. These British researchers argued that in many modern societies, young parents are viewed as less capable of fulfilling specific parenting functions and that this perception increases stigma and vulnerability to poor mental health. Another possible explanation they

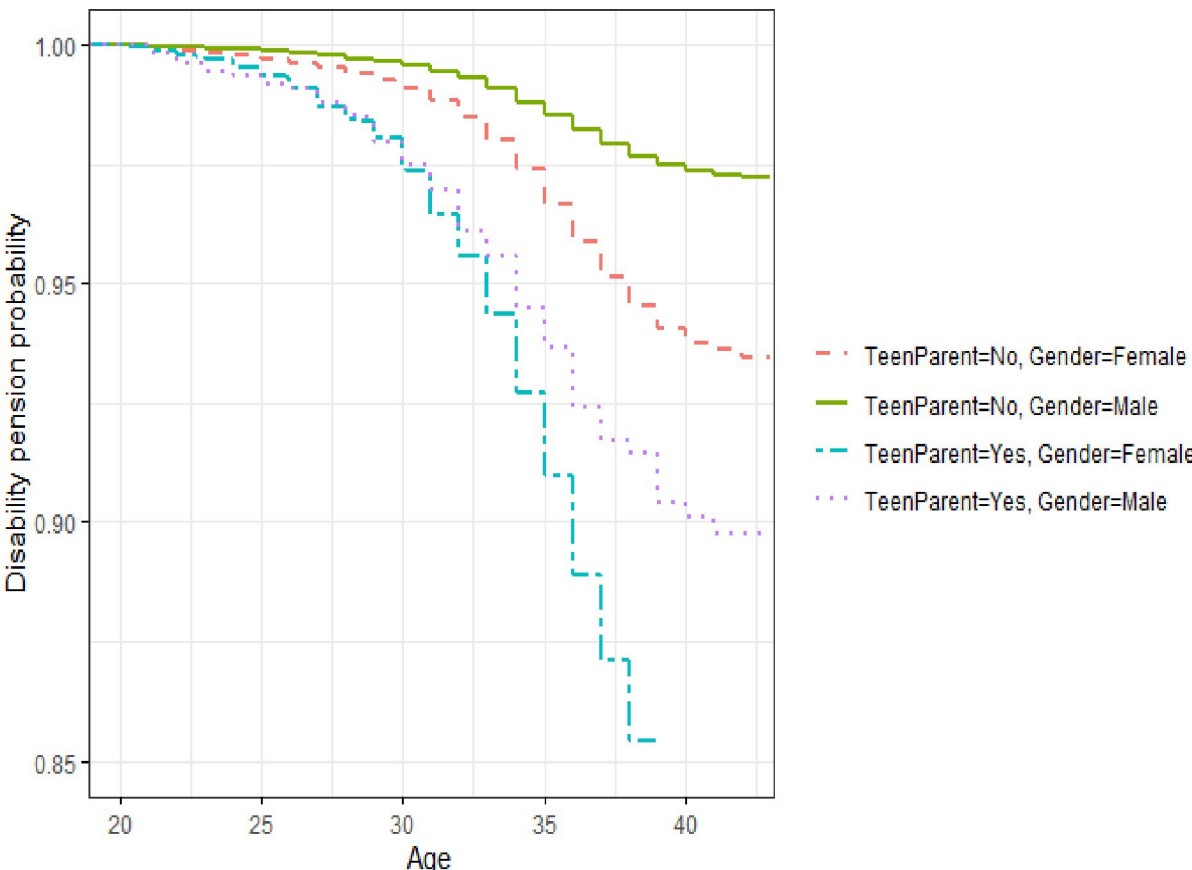

**Fig 1. Kaplan-Meier survival curves for age of receiving DP stratified by teenage parent status and sex.**

**Table 2. Cox regression results presented as hazard ratios (HR) and confidence intervals (CI) for the risk of receiving an early DP (at 20–42 years old), estimated for men and women.**

| | Men | | | Women | | |
|---|---|---|---|---|---|---|
| | Model 1 | Model 2 | Model 3 | Model 4 | Model 5 | Model 6 |
| Variable | HR (95% CI) | HR (95% CI) | HR (95% CI) | HR (95% CI) | HR (95% CI) | HR (95% CI) |
| **Teenage parent** | | | | | | |
| No | 1.0 | 1.0 | 1.0 | 1.0 | 1.0 | 1.0 |
| Yes | 2.4 (2.1–2.9)* | 2.3 (1.9–2.7)* | 2.3 (1.9–2.8)* | 2.3 (2.2–2.5)* | 2.4 (2.3–2.5) * | 2.4 (2.3–2.5)* |
| **Year of birth** | | | | | | |
| 1968 | | | 1.0 | | | 1.0 |
| 1969 | | | 0.9 (0.9–1.0)# | | | 0.9 (0.9–0.9)* |
| 1970 | | | 0.7 (0.7–0.8)* | | | 0.8 (0.8–0.8)* |
| **Father's education** | | | | | | |
| University | | 1.0 | 1.0 | | 1.0 | 1.0 |
| Secondary | | 1.4 (1.3–1.6)* | 1.5 (1.3–1.7)* | | 1.5 (1.4–1.6)* | 1.5 (1.4–1.6)* |
| Compulsory | | 1.9 (1.7–2.1)* | 1.9 (1.7–2.1)* | | 1.9 (1.8–2.1)* | 1.9 (1.8–2.0)* |
| Missing data | | 1.7 (1.5–1.7)* | 1.7 (1.5–1.9)* | | 1.2 (1.1–1.3)* | 1.2 (1.1–1.3)* |

*p-value <0.0001
#p-value = 0.24

offered was that teenage parenthood is often unplanned and tends to occur before a family unit is formed, which might imply that less social support is available. However, we think that it is also possible that the observed association between being a teenage parent and subsequent use of DP might be due to a selection effect, if there are other underlying factors such as drug use and mental health problems affecting both early parenthood and DP use.

It is evident from the Kaplan-Meier curves that teenage mothers used early DP more than teenage fathers. This finding is not surprising, since it is well established that women tend to use DP more often than men. This finding might suggest that teenage mothers are more exposed to health conditions that increase their risk of getting an early DP, despite Sweden's generous parental benefits including paid leave.

Our study found that persons whose fathers had had a low level of education were at increased risk of using an early DP. This finding supports previous studies that have linked poor socio-economic background to frequent use of DP [5, 37]. Having a low status on the social class gradient has been shown to be a strong predictor of high levels of sickness absence and is a recognized pathway to DP [37], and research has also consistently shown a relationship between low socio-economic status and high morbidity [38]. Poverty increases vulnerability to ill health due to material deprivation, unhealthy living conditions, and high levels of risk behaviors [38]. Exposure to high poverty levels during childhood and the teenage years may have significant effects on the health of young parents.

Our study findings suggest that teenage parenthood is associated with a subsequent early DP. We have not found any study that has investigated this association. We consider this to be an important finding given the high number of young people receiving DP not only in Sweden but also reported across several European countries [3, 39]. There is a need for more research to investigate factors associated with the early use of DP and the possible explanatory factors. Such findings might provide clues for public health interventions that could help in reducing teenage parenthood and promoting the health of young parents. It is worth noting that failure to enter the labor market or exiting the labor market early implies a long period spent outside the work environment, a factor that is linked to negative health, social and economic

consequences over the lifetime [39]. Moreover, data from Europe suggests that most people who start on DP tend to stay on it for a lifetime [39].

Our finding that teenage pregnancy is associated with an increased risk of early use of DP could be relevant to other Nordic countries that have a similar public support system. Considering that DP is a proxy for chronic disabling ill health, these results can apply to any context where teenage parenthood is common even in the absence of compensation schemes such as DP. Moreover, it can be argued that in the absence of financial compensation schemes, the effect of teenage pregnancy could be worse as ill health would be compounded by financial constraints. It is important to note that since teenage parenthood is rare in Sweden this might imply poor social support to teenage parents in Sweden compared to other contexts where teenage parenthood is common. The absence of social support might amplify the long-term social consequences of teenage parenthood [36]. We suggest further investigations into this relationship in other settings.

## Conclusions

Our study shows that teenage parents were significantly more likely to start using DP at the age of 20–42 years than their contemporaries who were not teenage parents. We observed a widening gap in the use of DP between ages 30–42, with the highest use among teenage mothers compared to teenage fathers and to non-teenage parents. The evidence of increased use of DP later in life among parents that had their first child at the age of 13–19 years suggests that there is increased chronic ill health among former teenage parents when compared to non-teenage parents which highlight possible health inequalities. It is important that future interventions aimed at reducing health inequalities give special consideration to young mothers and fathers. Additionally, future research should evaluate mechanisms that explain the association between teenage pregnancy and subsequent use of DP.

## Author Contributions

**Conceptualization:** Fredinah Namatovu, Lotta Vikström.

**Data curation:** Fredinah Namatovu.

**Formal analysis:** Fredinah Namatovu, Erling Häggström Gunfridsson.

**Funding acquisition:** Lotta Vikström.

**Investigation:** Fredinah Namatovu, Lotta Vikström.

**Methodology:** Fredinah Namatovu, Erling Häggström Gunfridsson.

**Validation:** Fredinah Namatovu.

**Writing – original draft:** Fredinah Namatovu, Lotta Vikström.

**Writing – review & editing:** Fredinah Namatovu, Erling Häggström Gunfridsson.

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
