## [Decision Letter · Decision Letter 0]

4 Nov 2022

PONE-D-22-17071Is teenage parenthood associated with early use of disability pension? Evidence from a longitudinal study

PLOS ONE

Dear Dr. Namatovu,

Thank you for submitting your manuscript to PLOS ONE. After careful consideration, we feel that it has merit but does not fully meet PLOS ONE’s publication criteria as it currently stands. Therefore, we invite you to submit a revised version of the manuscript that addresses the points raised during the review process.

We look forward to receiving your revised manuscript.

Kind regards,

Tae Kyoung Lee, Ph.D.

Academic Editor

PLOS ONE

Journal Requirements

Additional Editor Comments:

Dear Dr. Namatovu,

Thank you very much for submitting your manuscript "Is teenage parenthood associated with early use of disability pension? Evidence from a longitudinal study" for review and consideration for publication in PLOS ONE. I sincerely appreciate the opportunity to review the manuscript. I have now received the reviews of your manuscript and am able to make an editorial decision at this time.

Let me first sincerely apologize for the amount of review time for this submission. I am a firm believer in having at least 2 peer-reviews for convergence of perspectives, but a larger than average number of reviewers declined to review this work. This is not a reflection of the quality of the work; rather, it is a reflection of how much additional work reviewers of this journal have been presented with during the ongoing and uneven pandemic response.

The Reviewer and I have now commented on your paper. The reviews are appended to this email. As you will note, the reviewers found your manuscript to be quite interesting. However, at the same time, the reviewers also indicated that several quite significant conceptual and methodological/statistical problems exist in the manuscript, so I cannot accept the manuscript for publication in its present form. However, the reviewers and I conclude that it may be possible to address the points raised. I encourage you therefore to revise and resubmit the manuscript. I believe you will find the reviewers' comments clear.

Thank you for considering PLOS ONE as an outlet for your work.

Sincerely,

Tae Kyoung Lee, Ph.D.

Academic Editor

PLOS ONE.

Reviewers' comments:

Reviewer's Responses to Questions

**Comments to the Author**

1. Is the manuscript technically sound, and do the data support the conclusions?

Reviewer #1: Partly

Reviewer #2: Partly

2. Has the statistical analysis been performed appropriately and rigorously? 

Reviewer #1: Yes

Reviewer #2: No

3. Have the authors made all data underlying the findings in their manuscript fully available?

Reviewer #1: Yes

Reviewer #2: No

4. Is the manuscript presented in an intelligible fashion and written in standard English?

Reviewer #1: Yes

Reviewer #2: Yes

5. Review Comments to the Author

Reviewer #1: This is an interesting study that examines the longitudinal associations between teenage parenthood and early use of disability pension in Sweden. Below, I address issues/questions that emerged from my read of the paper.

[Introduction]

1. I was able to learn the longitudinal associations between teenage parenthood and early use of disability pension (DP) among Swedish. That’s advantage. However, I was wondering what contribution DP can make to society or what benefits it can have in their lives. Clarify whether getting DP is good or bad? If getting DP is good, why is this good?

[Result]

2. The study's findings that having the first child in 10s is associated with a higher There is not enough need for a relevance study to being a parent in your 10s and getting an early DP. (There is a lack of evidence as to why these studies should be done.)

3. Given that the author(s) used longitudinal data, I wonder if the findings was contaminated by confounders. Please consider socio-demographic variables (incomes, age, gender, parent education). It is obvious that your physical aging will come quickly if parenthood at an early age. However, readers may want to know what longitudinal changes of their physical health with the age of those who became parents in their 10s. If the data supports this, please provide evidence.

4. In the analyses, the author(s) checked comparison target the year of birth with the 3 year interval. Justify why you decided to use 3 years. Plus, there is a lack of information as to why only the father's education was adjusted and the mother's level of education was not taken into account.

[discussion]

5. I was wondering how this finding can be applied to other countries. Please discuss generalizability of the findings.

Reviewer #2: 

This is an interesting study conducted on early application and use of disability pensions (DP) for teenage parents. Below, I address some of the problems/questions that appeared when I read the paper.

Introduction

In the introduction, the study reported that the proportion of disability pensions (DP) in several European countries is increasing, and according to the WHO, one in seven people (14%) experience mental illness, accounting for 13% of the global burden caused by disease within this age group. It was said that this mental health had a negative effect, and the risk of higher health deterioration and socioeconomic consequences was high in teenage parents.

This high mental health problem can be identified as a strong predictor for using disability pension (DP) and hypothesized that becoming a teenage parent increases the risk of receiving disability pension (DP) in adulthood, and for the purpose of the study, it was examined whether there was a difference in the use of early disability pension (DP) between parents.

However, based on previous studies and introductions, it can be confirmed that there are already many mental health risks in adolescent parents, and as a result, it can be predicted that the probability of receiving a disability pension (DP) will be high.

Therefore, it is not well felt whether this content is an important explanatory element for research. If the research problem and hypothesis can be sufficiently confirmed based on the existing research, it will be necessary to derive and prove a more detailed research hypothesis.

I propose to mention the importance of research hypotheses in the discussion. In other words, why should readers know the finding of the hypothesis established by the researcher?

Results

The results reported in this study used analysis techniques through Kaplan-Meier curves and Cox provincial regression, but it was shown as an occurrence rate over simple time, and statistical significance verification of the results was not confirmed. There will be a need for solid information about the scientific validity of this report.

Discussion

This study reported that through discussion, the probability of teenage parents using disability pension (DP) and important results that teenage parents later adversely affect mental health were confirmed as implications.

If other research results or cohort comparison data that can support these implications were explained more abundantly, the validity of the results according to the research purpose and research hypothesis could be improved. However, since this research data is simple technical information, it is questionable whether it is information that can generalize the research results.

Conclusions

In conclusion, teenage parents are likely to start using the disability pension (DP) at the age of 20-42. However, it does not explain why teenage parents apply for a disability pension (DP) between the ages of 20-42.

In addition, it is based on Swedish cohort research data, and since Sweden's support policy is richer than other countries, the conclusion that other countries will have relatively larger problems is insufficient and it will need to be verified through comparison between countries in the future.

6. PLOS authors have the option to publish the peer review history of their article (what does this mean?). If published, this will include your full peer review and any attached files.

Reviewer #1: **Yes: **Minhee Lee

Reviewer #2: No

---

## [Author Response · Author response to Decision Letter 0]

1 Feb 2023

Dear Editor,

We would like to thank you as editor and the reviewers for the constructive feedback you have provided on our work. We have made most of the suggested revisions as requested and in a few instances where we haven’t made changes, we have endeavored to explain the reasoning behind our decision. Your feedback has helped make our manuscript stronger, we hope you will share this impression too. 

We hereby re-submit our manuscript, as a research article entitled, “Is teenage parenthood associated with early use of disability pension? Evidence from a longitudinal study” to be considered for publication in PLOS ONE. 

Below we address the comments raised by the editor.

1. We have ensured that our manuscript meets this journal’s requirements. 

2. Concerning data availability, we cannot make this data set publicly available according to Swedish ethical and legal restrictions in accordance with the Swedish Public Access to Information and Secrecy Act. However, this data can be made available to interested researchers by making a request directly to Statistics Sweden, a Swedish government agency responsible for official statistics in Sweden, to request access to the data, please contact information@scb.se, +46104795000. It should be noted that researchers interested in obtaining this data must also obtain ethical approval from the Swedish Central Ethical Review Board, contact, registrator@etikprovning.se, telephone: +46104750800. 

3. We could not upload the data due to restrictions from SEB and the ethical review board.

4. We have revised such that, the ethical statement only appears in the methods section.

Sincerely

Fredinah Namatovu

Response to the reviewers’ comments

 We would like to thank the reviewers for accepting to review our work and for providing valu

able feedback. We have addressed the issues raised chronologically and separately for each reviewer. 

Reviewer #1: 

This is an interesting study that examines the longitudinal associations between teenage parenthood and early use of disability pension in Sweden. Below, I address issues/questions that emerged from my read of the paper.

[Introduction]

1. I was able to learn the longitudinal associations between teenage parenthood and early use of disability pension (DP) among Swedish. That’s advantage. However, I was wondering what contribution DP can make to society or what benefits it can have in their lives. Clarify whether getting DP is good or bad? If getting DP is good, why is this good?

We thank the reviewer for this suggestion, we have revised the introduction section and added clarification on the importance of getting DP by adding the sentence below in the introduction section, lines 60-63: 

“Disability pension (DP) is considered an important part of the public support programs for people with disabilities. It is an essential social security program that acts as a salary replacement for people of working age with long-term health limitations preventing them from working [1].”

 [Result]

2. The study's findings that having the first child in 10s is associated with a higher There is not enough need for a relevance study to being a parent in your 10s and getting an early DP. (There is a lack of evidence as to why these studies should be done.)

Response: We consider our research question important, and our findings of valuable contribution given the fact that we found no existing literature examining the association between teenage parenthood and the subsequent risk of using DP. To make this straightforward for the reader we have added the statement below to the discussion section, lines 231-241

" So far, we have not found any study that has investigated the association between teenage parenthood and subsequent use of DP. Our study findings suggest that teenage parenthood is associated with a subsequent need for DP. There is a need for more research to investigate the causal mechanisms/ explanatory factors for this association. These findings might point to the possible public health approaches geared at improving the health and well-being of teenage parents as a way of preventing later chronic ill health that necessitates using DP early in life. Prevention is a key given that having DP implies exclusion from labor market participation at young age. Exiting the labour market early in life, implies long time spent outside the work environment which has been linked to negative health, social and economic outcomes over the life lifetime [23]. Moreover, data from Europe suggests that most people who start on DP tend to stay on it for a lifetime [23]. ”

3. Given that the author(s) used longitudinal data, I wonder if the findings was contaminated by confounders. Please consider socio-demographic variables (incomes, age, gender, parent education). It is obvious that your physical aging will come quickly if parenthood at an early age. However, readers may want to know what longitudinal changes of their physical health with the age of those who became parents in their 10s. If the data supports this, please provide evidence.

Response: Even though the confounders mentioned are important we did not include then in this analysis based on the following reasoning: a) Regarding physical aging, this was not possible to investigate due to data availability. We followed the birth cohort of 1968-1970 up to 2010 where data was available in our study, this means that the oldest individuals (born 1968) were 42 years making it impossible to draw conclusions regarding physical aging. Concerning other confounders we chose to include fathers’ education and birth year, the background variables measured at a clearly different time point to that when DP occurred to ensure tempolarity (that the exposure precedes the occurrence of the outcome, as suggested in the Bradford Hill criteria. We did not include income of the index person given DP is a salary replacement for those without a salary. 

4. In the analyses, the author(s) checked comparison target the year of birth with the 3-year interval. Justify why you decided to use 3 years. Plus, there is a lack of information as to why only the father's education was adjusted and the mother's level of education was not taken into account.

Response: Data availability was the main reason for using the three-year interval and father’s education. Since our data was available up to 2010, we wanted to have the population with the longest follow-up time, we chose the oldest individuals, those born between 1968-1970. We have revised the methods adding the statement below, lines 114-115. Regarding father’s education level, we have added the statement below on lines 134-136.

“These birth cohorts were chosen because they were the oldest in our data set and could provide sufficient follow-up duration.”

“Father’s level of education was used due to data availability, was grouped as the university, secondary, primary, and missing (for those who lacked data on education).”

 [discussion]

5. I was wondering how this finding can be applied to other countries. Please discuss generalizability of the findings.

Response: We think these results could be generalized to other contexts, such as Nordic countries that have a similar public financial compensation system for people with chronic ill health. Additionally, since DP is a proxy for chronic ill health, these findings could be applicable to any setting where teenage pregnancy is prevalent even in the absence of compensation schemes. We added a paragraph to elaborate further, lines 247-257.

“Our finding that teenage pregnancy is associated with an increased risk of early use of disability benefits could be relevant to other Nordic countries that have a similar public support system. However, given that DP is a proxy for chronic disabling ill health, these results can apply to any context where teenage parenthood is common even in the absence of compensation schemes such as DP. Moreover, it can be argued that in the absence of financial compensation schemes, the effect of teenage pregnancy could be worse as ill health would be compounded by financial constraints. However, since teenage parenthood is rare in Sweden this might imply poor social support to teenage parents in Sweden compared to other contexts where teenage parenthood is common. The absence of social support might amplify the long-term social consequences of teenage parenthood [37]. We suggest further investigations into this relationship in other settings.

Reviewer #2: REVIEW_MOON

This is an interesting study conducted on early application and use of disability pensions (DP) for teenage parents. Below, I address some of the problems/questions that appeared when I read the paper.

Introduction

In the introduction, the study reported that the proportion of disability pensions (DP) in several European countries is increasing, and according to the WHO, one in seven people (14%) experience mental illness, accounting for 13% of the global burden caused by disease within this age group. It was said that this mental health had a negative effect, and the risk of higher health deterioration and socioeconomic consequences was high in teenage parents.

This high mental health problem can be identified as a strong predictor for using a disability pension (DP) and hypothesized that becoming a teenage parent increases the risk of receiving disability pension (DP) in adulthood, and for the purpose of the study, it was examined whether there was a difference in the use of early disability pension (DP) between parents.

However, based on previous studies and introductions, it can be confirmed that there are already many mental health risks in adolescent parents, and as a result, it can be predicted that the probability of receiving a disability pension (DP) will be high.

Therefore, it is not well felt whether this content is an important explanatory element for research. If the research problem and hypothesis can be sufficiently confirmed based on the existing research, it will be necessary to derive and prove a more detailed research hypothesis. The hypothesis cannot be sufficiently confirmed from existing studies. I propose to mention the importance of research hypotheses in the discussion. In other words, why should readers know the finding of the hypothesis established by the researcher?

Response: We thank the reviewer for this reflection. It is true that there are several studies already showing that teenage pregnancy is associated with increased mental health problems. We use this argument as a starting point, but do not investigate mental health problems, instead, we look at DP. Based on our literature search, we did not find any study that has investigated the link between teenage parenthood and subsequent use of DP. We think this is an interesting research question, given the increasing challenge of the early use of DP in Europe. To make this more evident, we have revised the discussion section, adding a paragraph below on lines 238-250.

“Our study findings suggest that teenage parenthood is associated with a subsequent need for DP, so far, we have not found any study that has investigated this association. This is an important finding considering the fact that the number of young people receiving DP has rapidly increased not only in Sweden but also across Europe [40]. There is a need for more research to investigate factors associated with the early use of disability pensions and the possible explanatory factors. Such findings might provide cues for possible public health interventions geared at reducing teenage parenthood and support for young parents to achieve quality health and prevent chronic ill health that necessitates the use of DP early in life. Failure to enter the labour marker or exiting the labor market early due to chronic ill-health implies a long period spent outside the work environment, a factor that has been linked to negative health, social and economic consequences over the lifetime [40]. Moreover, data from Europe suggests that most people who start on DP tend to stay on it for a lifetime [40].” 

Results

The results reported in this study used analysis techniques through Kaplan-Meier curves and Cox provincial regression, but it was shown as an occurrence rate over simple time, and statistical significance verification of the results was not confirmed. There will be a need for solid information about the scientific validity of this report. 

Response: We performed a Cox proportional hazard regression analysis, a type of analysis strongly recommended for our type of data, the longitudinal data with long follow-up duration. In the previous version, we reported both independent and adjusted associations as hazard ratios (HR) with 95% confidence intervals. All results were statistically significant based on the confidence intervals and p-values. We have added information on the p-values in table 2, line 187, on line 188 we give the p-values below:

“p-value <0.0001, p-value =0.24”

Discussion 

This study reported that through discussion, the probability of teenage parents using disability pension (DP) and important results that teenage parents later adversely affect mental health was confirmed as implications. If other research results or cohort comparison data that can support these implications were explained more abundantly, the validity of the results according to the research purpose and research hypothesis could be improved. However, since this research data is simple technical information, it is questionable whether it is information that can generalize the research results.

Response: We have made the revision below, similar to our response to the first comment, see lines 238-250. 

“Our study findings suggest that teenage parenthood is associated with a subsequent need for DP, so far, we have not found any study that has investigated this association. We think this is an important research contribution considering the fact that the number of young people receiving DP has rapidly increased not only in Sweden but also across Europe [40]. There is a need for more research to investigate factors associated with the early use of disability pensions and the possible explanatory factors. Such findings might provide cues for possible public health interventions geared at reducing teenage parenthood and support for young parents to achieve quality health and prevent chronic ill health that necessitates the use of DP early in life. Failure to enter the labour marker or exiting the labor market early due to chronic ill-health implies a long period spent outside the work environment, a factor that has been linked to negative health, social and economic consequences over the lifetime [40]. Moreover, data from Europe suggests that most people who start on DP tend to stay on it for a lifetime [40].” 

Conclusions

In conclusion, teenage parents are likely to start using the disability pension (DP) at the age of 20-42. However, it does not explain why teenage parents apply for a disability pension (DP) between the ages of 20-42. In addition, it is based on Swedish cohort research data, and since Sweden's support policy is richer than other countries, the conclusion that other countries will have relatively larger problems is insufficient and it will need to be verified through comparison between countries in the future.

Response: We have revised the conclusion, but also added a paragraph in the discussion to expound in this point.

See the revision below on lines 252-262

“Our finding that teenage pregnancy is associated with an increased risk of early use of disability benefits could be relevant to other contexts, that have similar public support systems, like the Nordic countries. Given that DP is a proxy for chronic disabling ill health, these results can apply to other contexts where teenage parenthood is common but without a compensations scheme such as DP. Moreover, in the absence of such financial compensation schemes, the effect of teenage pregnancy could be worse as ill health could be compounded by financial constraints. However, since teenage parenthood is rare in Sweden this might imply poor social support to teenage parents compared to other contexts where teenage parenthood is common. This could mean that the related long-term social consequences of teenage parenthood might be more severe in Sweden due to a lack of social [37]. To draw firm conclusions, we suggest further investigations into this relationship in other settings”.

---

## [Editor Report · Decision Letter 1]

2 Jun 2023

Is teenage parenthood associated with early use of disability pension? Evidence from a longitudinal study

PONE-D-22-17071R1

Dear Dr. Namatovu,

We’re pleased to inform you that your manuscript has been judged scientifically suitable for publication and will be formally accepted for publication once it meets all outstanding technical requirements.

Kind regards,

Janet E Rosenbaum, Ph.D.

Academic Editor

PLOS ONE
---

## [Editor Report · Acceptance letter]

5 Jun 2023

PONE-D-22-17071R1 

Is teenage parenthood associated with early use of disability pension? Evidence from a longitudinal study 

Dear Dr. Namatovu:

I'm pleased to inform you that your manuscript has been deemed suitable for publication in PLOS ONE. Congratulations! Your manuscript is now with our production department. 

Kind regards, 

on behalf of

Dr. Janet E Rosenbaum 

Academic Editor

PLOS ONE